# From Drug Discovery to Drug Approval: A Comprehensive Review of the Pharmacogenomics Status Quo with a Special Focus on Egypt

**DOI:** 10.3390/ph17070881

**Published:** 2024-07-03

**Authors:** Fadya M. Elgarhy, Abdallah Borham, Noha Alziny, Khlood R. AbdElaal, Mahmoud Shuaib, Abobaker Salem Musaibah, Mohamed Ali Hussein, Anwar Abdelnaser

**Affiliations:** 1Institute of Global Health and Human Ecology, School of Sciences and Engineering, The American University, Cairo 11835, Egypt; fadya_elgarhy@aucegypt.edu (F.M.E.); abdallahborham@aucegypt.edu (A.B.); nohaalziny@aucegypt.edu (N.A.); mahmoudshuaib@aucegypt.edu (M.S.); abobaker@aucegypt.edu (A.S.M.); mohamed_hussein@aucegypt.edu (M.A.H.); 2Egypt Center for Research and Regenerative Medicine (ECRRM), Cairo 4435121, Egypt; 3Graduate Program of Biotechnology, School of Sciences and Engineering, The American University, Cairo 11835, Egypt; khlood.ramadan@aucegypt.edu

**Keywords:** PGx, pharmacogenomics, regulatory, drug approval, personalized medicine

## Abstract

Pharmacogenomics (PGx) is the hope for the full optimization of drug therapy while minimizing the accompanying adverse drug events that cost billions of dollars annually. Since years before the century, it has been known that inter-individual variations contribute to differences in specific drug responses. It is the bridge to what is well-known today as “personalized medicine”. Addressing the drug’s pharmacokinetics and pharmacodynamics is one of the features of this science, owing to patient characteristics that vary on so many occasions. Mainly in the liver parenchymal cells, intricate interactions between the drug molecules and enzymes family of so-called “Cytochrome P450” occur which hugely affects how the body will react to the drug in terms of metabolism, efficacy, and safety. Single nucleotide polymorphisms, once validated for a transparent and credible clinical utility, can be used to guide and ensure the succession of the pharmacotherapy plan. Novel tools of pharmacoeconomics science are utilized extensively to assess cost-effective pharmacogenes preceding the translation to the bedside. Drug development and discovery incorporate a drug-gene perspective and save more resources. Regulations and laws shaping the clinical PGx practice can be misconceived; however, these pre-/post approval processes ensure the product’s safety and efficacy. National and international regulatory agencies seek guidance on maintaining conduct in PGx practice. In this patient-centric era, social and legal considerations manifest in a way that makes them unavoidable, involving patients and other stakeholders in a deliberate journey toward utmost patient well-being. In this comprehensive review, we contemporarily addressed the scientific leaps in PGx, along with various challenges that face the proper implementation of personalized medicine in Egypt. These informative insights were drawn to serve what the Egyptian population, in particular, would benefit from in terms of knowledge and know-how while maintaining the latest global trends. Moreover, this review is the first to discuss various modalities and challenges faced in Egypt regarding PGx, which we believe could be used as a pilot piece of literature for future studies locally, regionally, and internationally.

## 1. Introduction

The Human Genome Project was an eye-opening trek into the massive potential of genomic research. Pharmacogenomics (PGx) is a comprehensive approach that concerns the interaction between the human genome and exogenous substances. One of its main pillars is the accuracy of therapy and the safety of the drug on individuals. PGx studies the variations in the whole human genome and how they might affect drug response. However, pharmacogenetics is the daughter of PGx and can be defined as ‘the study of changes in a single gene and how it can affect drug activity’ [1]. The birth of PGx was upon the realization that genetic variations are the main reason for inconsistencies in drug responses. Because of how crucial it is to anticipate the body’s response to a certain drug, taking PGx one step backward to drug design is invaluable. Variation in responsiveness pushes us to harness both individual and population-level genomics (Figure 1). 

Currently, the capacity of PGx, namely the diagnosis of critical variants capable of drug toxicities, is rather underused. The fact that PGx is needed in therapy design and optimization is undeniable. Two roadblocks to channeling PGx from early biomedical research to the bedside are manifesting cost-effectiveness and fostering clinical implementation. In the quest to make PGx a fundamental aspect of pharmacotherapy and the management of various ailments, multiple stakeholders must be recognized. Applying genomic advancements is both time-saving and money-saving for pharmaceutical giants, yet they prefer to envision an unsegmented market. 

On the other hand, with the influx of evidence for clinical effectiveness and availability of the required kits, it is of great necessity that clinicians be fully updated with PGx therapeutic value and also its limitations in patient care [2]. Not only does drug metabolism occur in various parts of the body, but the liver holds 70% of the body’s share in drug metabolism. The drug clearance process consists of two phases, in which a superfamily of enzymes, ‘cytochrome P450’, are in charge for more than 75% of its phase 1 metabolism. The cytochrome P450 (CYP 450) enzymes family executes drug bioactivation and/or detoxification; therefore, harnessing genetic mutations in the genes encoding them is of serious concern. They encode drug-metabolizing enzymes, transporters, drug targets, HLA alleles, drug efficacy prediction, or adverse events would be of great benefit if they are a reliable PGx biomarker. Genetic polymorphisms, the most crucial factor, account for 20–30% of the inter-individual differences in treatment response.

Disease heterogeneity, environmental, and other inherent patient characteristics such as ethnicity, age, or weight, among others, are collectively contributing to varying drug responses. Throughout early drug development studies, pharmacokinetics (PK) and pharmacodynamics (PD) investigations explicitly describe how intricate individual differences are. Without a doubt, drug success is achieved by weighing benefit against risk, which is composed of this trio: efficacy, safety, and the right dosage [3]. Personalized medicine, in other words, the 4Ps medicine, is a concept that aims to achieve multidimensional human health: Personalized, Predictive, Preventive, and Participatory [4]. 

This review article aims to provide a holistic approach to reviewing the current status of PGx in Egypt, highlighting the national Egyptian Genomic projects. Those projects will empower policymakers and the government to implement PGx in drug development and regulation, paving the way for applying the personalized health concept in Egypt. Consequently, this collective effort will be reflected in the clinical practice, and fundamentally, the Egyptian patient will get the maximum benefit from the approved drugs seamlessly. 

## 2. The Current Status of PGx

Following the completion of the Human Genome Project in 2003, understanding how human sequence variants impacted drug research and effectiveness prediction paved the way for more personalized approaches in drug selection and prescribing. Understanding how interindividual variability could affect treatment response—that is, why some individuals who have the same condition respond to the same medication differently—is made more comprehensible. It also suggests the possibility of treatment failure and the development of adverse drug reactions (ADRs). 

Topić et al. revealed that only one-half of the prescribed medications have the anticipated treatment efficacy. Another startling statistic is that the ADRs are responsible for approximately 7% of all hospital admissions and nearly 20% of readmissions, making them the fourth most common cause of mortality, with an estimated USD 136 billion in yearly costs. The authors demonstrate the projected therapeutic effectiveness of just half of the recommended medications. Intriguingly, genetic factors may be responsible for up to 20% of all ADRs that are reported [5].

PGx has made significant strides in understanding how genetic variations influence an individual’s medication response. With continuous research in the field and its vast application, it identifies genetic markers associated with drug metabolism, efficacy, and adverse reactions [6]. The idea of personalized healthcare has become increasingly popular in modern medicine, partly due to developments in PGx. Within personalized medicine, comprehending the interplay between drugs and genes is essential for customizing treatment regimens for specific patients, maximizing effectiveness, and reducing side effects. PGx redefines personalized medicine by allowing medical professionals to anticipate which drugs will work best and be safest for individual patients. 

Genetic variations majorly impact drug metabolism, efficacy, and toxicity, which can influence treatment outcomes. For instance, differences in the *CYP2D6* gene can change how many antidepressants and antipsychotic drugs are metabolized, which can change the effectiveness of the medication and increase the chance of side effects. 

This section will discuss some of the critical genes associated with drug-gene interactions, preview the role of genetic polymorphisms in determining the fate of the drugs in the human body, and later illustrate the inconsistencies of PGx drug labels. It will illustrate the pharmacogenes based on the prevalence of genetic variations that have a noticeable impact on drug metabolism, and the clinical significance of those gene variations affecting treatment safety and effectiveness. 

Drugs exert their biological action by passing through the cell membranes from the application site. Then, it is transported to cell compartments, followed by a biotransformation process, and finally eliminated from the body. Any gene variation involved in this process could influence expression which subsequently affects the drug response. The major players in the drug journey in the body are PK and PD, simply what the body does to the drug and what the drug does to the body, respectively. These factors determine the drug’s response and fate in the human body. That is one of the reasons behind adopting the term PGx more than pharmacogenetics, as it is a multi-gene interaction involving the drug response, leading to the necessity of the systematic study of drug–gene interactions. Here, we will examine some essential genes and variations influencing drug response.

The CYP450 family of enzymes is one of the widespread enzyme families involved in drug-metabolizing enzymes that are essential for the biotransformation of any xenobiotic that enters the human body, particularly CYP2D6, CYP2C9, and CYP2C19 [7]. Genetic polymorphism in one of the PK parameters, including absorption, distribution, metabolism, and excretion (ADME), subsequently affects the concentration of the drug or its metabolites in the body resulting in variation in drug response. The human genome’s sequencing process identified 115 *CYP450* genes, of which only 57 are active and encode distinct enzymes, while the remaining 115 genes are pseudogenes that do not function but cause errors in polymorphism detection techniques [5].

CYP2C19 is quite similar to CYP2C9 as it is accountable for the metabolism of similar drug groups: anticonvulsants, proton pump inhibitors, e.g., omeprazole, antidepressants, e.g., citalopram, and antiplatelet drugs, such as clopidogrel. CYP2C19 is extensively studied with clopidogrel therapeutic failure, as clopidogrel is a prodrug that needs to be activated by CYP2C19. While the presence of the *CYP2C19*17* allele is associated with the risk of bleeding in these patients, the presence of *CYP2C19* alleles **2*, **3*, **4*, **5*, **6*, and **8* leads to reduced antiplatelet activity as the drug is not converted into the active form [5]. A study conducted in the Egyptian population shows that the **2* and **3* alleles are the most prevalent allelic variations in *CYP2C19* and are regarded as loss-of-function alleles. Individuals with these alleles are considered aberrant clopidogrel metabolizers since they are more resistant to the medication, with 15.3% and 1.2% having the two alleles, respectively [7].

### 2.1. CYP2D6 and Pimozide

CYP2D6 is an enzyme that is in charge of the metabolism of certain antidepressants, neuroleptics, antiarrhythmics, and β-adrenergic blockers, and it is mainly found in the liver and central nervous system, which also plays a vital role in the metabolism of dopamine precursors, neurosteroids, and serotonin as endogenous substrates [8]. According to the Pharmacogenomics Knowledge Base (PharmGKB), it is very polymorphic and contributes to the metabolism of up to 25% of medications [9]. Further, CYP2D6 is one of the enzymes responsible for clinically significant toxicity related to codeine [10], tramadol [11], antidepressants, and opioids.

Pimozide is an antipsychotic drug that has shown significance in pharmacogenetic testing, in which regulatory officials state that pimozide should undergo genetic testing before being prescribed. The Food and Drug Administration (FDA) label for Pimozide specifies that adults with *CYP2D6* genotyping should be tested for genetic variations at dosages of more than 4 mg/day. Therefore, it is highly recommended to be tested in psychiatric clinics as part of regular testing, highly beneficial to patients on antidepressants and psychotic drugs [12]. 

### 2.2. CYP2C9 with Warfarin and Phenytoin

CYP2C9 is one of the most significant P450 family members, and it has remarkable correlations (>82%) with DNA and protein sequences. While CYP2C8 and CYP2C19 are expressed at two and ten times lower levels than CYP3A4, respectively, CYP2C9 is the most expressed member, expressed at comparable or even higher protein levels [13]. While CYP2D6 has an essential role in opioid-related toxicity, CYP2C9 is the one responsible for the metabolism of the classes of anticonvulsants, proton pump inhibitors, antidepressants, and antiplatelet medications like one of the most well-studied medications, warfarin [5]. CYP2C9 is well-known for its implications with warfarin, the oral vitamin K antagonist anticoagulant, a drug that is known for its treatment challenges because of its narrow therapeutic range and intricate dose response.

In a recent study, individuals with the *CYP2C9*2* or **3* LoF, when treated with clopidogrel for acute coronary syndrome, did not have a higher risk of thrombotic events than those without the allele [14]. It also directly affects the antiepileptic drug phenytoin. It also has a narrow therapeutic range, and it takes a lot of effort and money to determine the right, safe, and effective dose. Thus, it is advised to screen for CYP2C9 polymorphisms before beginning the medication [5].

Interestingly, despite all the research conducted on *CYP2C9* and its various alleles which implicated several anticoagulants and non-steroidal anti-inflammatory drugs that may raise a challenge among clinicians to prescribe the appropriate drug, the Clinical Pharmacogenetics Implementation Consortium (CPIC) guidelines have not been updated since 2017. That indicates the complexity of the interactions between alleles and other contributing factors. Noticeably, *CYP2C9*2* and **3* are the tested alleles with the most warfarin and phenytoin in the tested patient, indicating how critical those alleles are in their metabolism or activation. 

### 2.3. CYP2C9 and VKORC1 with Warfarin

The main genetic factor responsible for variation in warfarin dose requirements is a polymorphism in the vitamin K epoxide reductase complex 1 (*VKORC1*) gene [15]. Vitamin K epoxy reductase, a crucial enzyme in the vitamin K cycle that is encoded by the *VKORC1* gene, is responsible for the transforming of vitamin K epoxide from an inactive form into a physiologically active reduced form. This reduced form subsequently activates coagulation factors II, VII, IX, and X.

Genetic polymorphisms *CYP2C9* (~10%) and *VKORC1* (~25%) can explain patients’ interindividual variability when warfarin is administered. Although, non-genetic factors, such as age, weight, sex, and BMI, account for about 20% of the total variability in the therapeutic response [5].

### 2.4. SLCO1B1 and Statins

*SLCOB1* is a gene-encoding anion transporting polypeptide 1B1 (OATP1B1) protein, which is present mainly in the liver and acts as a transporter that carries substances from the blood into the liver to be eliminated. It is responsible for transporting various xenobiotics, and one of the most studied substances is statins [16]. Statins are lipid-lowering medications that are widely used and considered safe medications; however, they can induce myopathy ranging from mild myalgia to rare and potentially fatal rhabdomyolysis. It is referred to as statin-induced myotoxicity. The OATP1B1, a substrate for statins, is encoded by the *SLCO1B1* gene. A polymorphism in *SLCO1B1* (*SLCO1B1* 521T>C SNP) has been shown to dramatically reduce the hepatic absorption of statins and increase their systemic exposure. A genome-wide association studies (GWAS) analysis has found a significant association between simvastatin-induced myopathy and the *SLCO1B1* 521T>C SNP [17]. 

There is substantial evidence linking rs4149056 to muscle toxicity caused by simvastatin; however, there is little evidence linking this polymorphism to other statin safety. Prior research assessed the safety of atorvastatin in people with the rs4149056 mutation [16]. CPIC guidelines in 2014 recommend an alternative statin other than simvastatin if the patient with a C allele at rs4149056 does not achieve optimal LDL cholesterol-lowering efficacy.

### 2.5. ADRB1 and 2 Receptors with Cardiovascular Diseases (CVDs)

The adreno receptor beta 1 (*ADRB1*) gene and the *ADRB2* gene encode the beta-1 and 2-adrenergic receptors. Both receptors are G-coupled receptors and are expressed mainly in cardiac tissues. Both receptors are essential for maintaining normal cardiac function. Few polymorphisms associated with those receptors have been reported in heart diseases. For instance, genetic heterogeneity in the beta 1-AR, at position 389, influences the effect of beta-AR blockers in vivo as well as the functional response to agonists [18]. Vasodilation, bronchodilation, heart rate, and contractility are all crucially regulated by beta 2-AR. It is reported that positions 16 and 27 do not functionally affect heart rate or contractility. On the other hand, the Arg16Arg–Gln27Gln-beta2AR in bronchial and vascular smooth muscle is particularly vulnerable to desensitization caused by agonists.

Both beta-1- and beta-2-AR are changed in heart failure; these changes are likely also influenced by the genetic variations of the beta-1 and beta-2-AR. However, beta AR single-nucleotide polymorphisms (SNPs) are not found to be causing cardiac diseases; instead, they indicate the gene’s susceptibility to agonists and antagonists that determine the patient’s medication response [18].

### 2.6. Other Involving Enzymes: Thiopurine S-Methyltransferase (TPMT)

TPMT is one of the major enzymes involved in phase II drug metabolism and responsible for facilitating its inactivation or excretion. Azathioprine and 6-mercaptopurine are among the drugs in which TPMT plays a noticeable role in their metabolism as it breaks down their thiopurine ring. Alleles **3A* and **3C*, **3B*, and **2* are linked to non-functional or low enzyme activity TPMT alleles. Hence, pharmacogenetic testing of the TPMT **2*, **3A*, **3B*, **3C*, and **4* alleles with azathioprine-treated patients is recommended due to the hematotoxicity of the low TPMT activity [19]. Considering all previous examples, PGx’s current state elucidates the field’s continuous difficulties and incredible advancements. Genetic data incorporation into medication development and clinical practice has improved the knowledge of individual differences in drug response, resulting in more individualized and efficient therapies. A few critical challenges must be resolved to fully achieve PGx’s promise. Ultimately, integrating PGx into clinical practice will require a thorough understanding and resolution of pharmacological reactions’ genetic complexity and diversity.

### 2.7. The Effect of Genetic Polymorphism on Anticancer and Antiviral Treatment

Several anticancer drugs are associated with severe, or even life-threatening, side effects. For instance, targeted therapeutic agents such as tyrosine kinase inhibitors (TKIs), including imatinib, dasatinib, and nilotinib, are extensively metabolized by CYP450 enzymes; therefore, genetic polymorphisms in those genes greatly impact treatment response. Imatinib is the first generation TKI used in treating *BCR-ABL* acute myeloid leukemia and chronic myeloid leukemia (CML). It is extensively metabolized by CYP450 enzymes, especially CYP3A4 and other enzymes such as CYP2C8, CYP3A5, and CYP2D6, with less contribution in the metabolizing that converts Imatinib to more active metabolite. One study has reported that in the expression in CML, patients who achieve a complete response have higher expression levels of CYP3A4 and CYP3A5 compared to partial responders, insinuating the rule of CYP450 in drug response [20]. Thus, examining genetic variants involved in imatinib metabolism for personalization of the treatment, including the *CYP3A5*3* (GG) genotype, which has been reported to be associated with a higher level of plasma concentration but not clinical outcomes, and the *CYP2B6* 15631GG/TT genotype which is associated with a complete hematological response and primary cytogenetic while patients with the *15631GG/GT* genotype achieve a higher complete cytological response and are more susceptible to developing side effects [21,22]. Nilotinib is a common drug used in imatinib-resistant or imatinib-intolerant CML, and it is recommended to test for the genetic variant associated with nilotinib-induced hyperbilirubinemia, such as *UGT1A1*28* and *UGT1A1*6*, ahead of nilotinib administration [23,24,25,26]. In addition, several genetic variants are associated with drug transport genes, such as *ABCB 11263* (TT), (CT/TT); *ABCB 2677* (TT/TA), (G); *ABCB 3435* (TT), (CC), *ABCG 234* (G>A), *ABCG 421* (C>A), *SLC22A1* rs3798168, rs628031, *IVS7+ 850* (C>T), *SLC22A1 480* (C>G), *SLC22A1 401* (G/A), *SLC22A4* rs1050152, *SLCO1A2 361* (GG), *SLCO1B3 334* (GG), (TT); *SLCO1B3 699* (GG), *ABCA3 4548–91* (CC/CA), which are associated with mediating drug resistance, increased drug clearance, and decreased treatment outcomes of TKIs in CML (reviewed comprehensively in reference [23]). In addition, there are several other anticancer-targeted therapies such as the vascular endothelial growth factor receptor (VEGFR) and various monoclonal antibodies (reviewed in reference [27]). Studies demonstrate that in triple-negative breast cancer, patients with overexpression of the *CREB3L1* gene treated with doxorubicin achieve a better therapeutic response, making it an interesting biomarker to examine doxorubicin response [28,29].

Despite doxorubicin’s wide range of uses in breast cancer, acute lymphoblastic leukemia, and other malignancy treatments, it has a variety of harmful effects, with cardiotoxicity being the most well-known and thoroughly researched side effect due to polymorphism and its dosage [30]. Several studies have been carried out to investigate the genetic contribution to this toxicity. The expression of genes related to inflammatory and immune dysregulation and the presence of human leukocyte antigen (HLA) single nucleotide polymorphism predisposes patients to cardiotoxicity [31]. In acute lymphoblastic leukemia, acute and chronic cardiotoxicity has been linked to polymorphisms in components of the *NAD(P)H* [32] and *NOS3* genes [33].

Furthermore, antiviral drugs are also subjected to CYP450 which potentially affects drug response as well as the associated adverse events. In a review article conducted by Owen et al. [34] the authors comprehensively examined the associations between different antiviral medications and genetic variants in CYP450 and drug transporters. For example, polymorphism on *CYP2B6* results in higher concentration of efavirenz in plasma which consequently resulted in increased central nervous system toxicity. However, a discrepancy has been associated with ABCB1 genetics polymorphism, and with the efficacy of abacavir, the association with hypersensitivity reaction is well established. In addition, the abacavir hypersensitivity reaction is strongly associated with the haplotype *HLA-B*5701*, *HLA-DR7*, and *HLADQ3* [28,29]. 

## 3. Current Status in Egypt

Egyptian healthcare sector leaders and policymakers are shifting from a uniform approach, which is well-known as “one-size-fits-all”, to a more tailored approach, which thrives on the advancements in medical genetics and genomics studies nationwide. This transition highlights the importance of having a thorough understanding of the Egyptian genome and the related disorders to guide the implementation of effective preventive, diagnostic, and counseling measures for common genetic conditions in Egypt [35]. 

The scientific and academic community is working diligently to acquire more data to draw significant conclusions about the highly diverse population of Egypt. Current genomic reference databases do not adequately represent the global human population, which presents challenges in interpreting variants, especially in under-represented groups like the North African inhabitants including Egypt. In that context, the Egyptian Collaborative Cardiac Genomics (ECCO-GEN) project started an investigation involving 1000 individuals, “Healthy Volunteers”, who were free of CVDs to gather individual-level genetic and phenotypic data to shape the future of CVDs research and, hence, diagnostic and treatment strategies. This study revealed the findings of the first 391 healthy volunteers recruited to establish a pilot phenotype control cohort. Each participant underwent a comprehensive clinical assessment and genetic sequencing using a targeted panel including 174 genes associated with inherited cardiac conditions. A total of 1262 variants were detected across 27 cardiomyopathy genes, with 15.1% of variants not accounted for in the current genetic reference databases, such as gnomAD and Great Middle Eastern Variome. The project was focused on characterizing the Egyptian genetic makeup, which has not been extensively studied [36].

Similarly, a study investigating the biallelic *ALKBH8* variant in neurodevelopmental disorders was conducted on a family of Egyptian descendants. ALKBH8 is a methyltransferase that modifies tRNA to preserve codon recognition and prevent translation errors. Two homozygous truncating variants in *ALKBH8* have been previously linked to intellectual development disorder and MRT71 syndrome in two large, unrelated Saudi Arabian families (MIM# 618504). The study identified a novel homozygous frame-shift variant in the final exon of *ALKBH8* in a third family of Egyptian descent, leading to global developmental delay and intellectual disability. Consequently, this study expands the phenotypic and genotypic spectrum of MRT71 syndrome and confirms ALKBH8 as a promising neurodevelopmental disease biomarker [37].

On the same yet more advanced approach, a study conducted on 537 Egyptian patients and 883 controls is considered the first GWAS to identify novel susceptibility loci in systemic lupus erythematosus (SLE). The authors identified a novel significant locus near *IRS1/miR-5702* and eight suggestive loci. The study replicated 97 previously known loci, with *ITGAM*, *DEF6-PPARD*, and *IRF5* as the top three. SNPs from four potential loci corresponding with lead SNPs were linked to different gene expressions. These loci are implicated in crucial pathways in SLE and nephritis. Yet, small sample sizes impacted the statistical power and generalizability of the findings. Therefore, more studies with large sample sizes are warranted [38]. 

Furthermore, another study recruited 110 healthy individuals and used genome sequencing along with assembly techniques to construct an Egyptian genome reference dataset. The dataset included 19,758,992 single nucleotide variants, 121,141 structural variants, and mitochondrial haplogroups, revealing four major genetic ancestry components in Egyptians and identified 1198 Egyptian population-specific variants, including 49 novel variants. In addition, the authors highlighted haplotype differences between Egyptians and Europeans, which could impact a crucial genetic risk assessment, and emphasized the differences in allele frequencies and linkage disequilibrium between Egyptians and Europeans [39]. These differences may affect European ancestry-based genetic disease risk transferability and polygenic scores to the Egyptian population. 

Amin et al. have found that patients carrying the allelic polymorphism rs205764 on linc00513—which has been reported as a novel regulator of the type 1 interferon signaling pathway in multiple sclerosis patients—significantly responded more to fingolimod compared to patients carrying the major allele [40]. In addition, Donkol et al. have studied the drug response in 100 Egyptian patients treated with warfarin and revealed that patients carrying the rs9934438 SNP located in the *VKORC1* gene of a heterozygous (A/G) genotype required a lower daily warfarin dose than the other variants (*p* = 0.006) which is not common in a highly racially mixed population, such as the Egyptian population [41]. In another study, genotyping of *VKORC1A* and *CYP2C9* genetic polymorphism was performed to identify the optimum warfarin dose. It was evident that patients carrying the *VKORC1* homozygous (AA) genotype should have lower doses of warfarin. For CYP2C9, the patients carrying the *CYP2C91*1* allele should take higher doses of warfarin, while patients carrying *CYP2C93*3* should take lower doses. Another study conducted on warfarin explored the effect of other genes on drug response. Egyptian patients carrying genetic polymorphisms of both *MDR1* and *EPHX1* should administer a higher dose of warfarin. However, the dose alteration is not recommended for patients carrying an SNP in only one gene. On the contrary, for the patients carrying the wild-type *EPHX1* HH and *MDR1* TT, the warfarin dose should be decreased in comparison with *EPHX1* heterozygous and homozygous (HR and HH) variants and *MDR1* variants (CC and CT) subjects [42]. 

In patients with hypercholesterolemia, Sabokbar et al. investigated the influence of polymorphisms in two genes, *MDR1* and *SLOC1B1*: the former encodes for an influx transporter protein while the latter codes for an efflux transporter protein, respectively, on the drug response of atorvastatin. The *MDR1* C3435TT homozygous variant was associated with higher HDL-C levels in baseline and post-treatment with atorvastatin compared with the other two variants (CC and wild-type). A gender–gene interaction was seen in all genetic variants of *SLCO1B1* A388G related to statin therapeutic response and metabolism. The *SLCO1B1* gene is located on chromosome 12p12.2, and non-synonymous SNPs have been identified within this gene, which may affect the transporter activity [43]. Collectively, most of these local data are generated with domestic studies validating global PGx data. Therefore, systematically reviewing observations and interventions in the Egyptian population would be a recourse for guiding medical practice with cumulative, credible evidence. In addition, the rs464637 in *CYP3A4* has been involved with cyclosporine, used as an immunosuppressor in renal transplantation and other immunological diseases. It is also reported that there is an association between *HLA-A1* and susceptibility to viral clearance following PEG-IFN/RBV therapy in the Egyptian population. It is also suggested that *IL28B* polymorphisms can be used as pre-treatment biomarkers in predicting susceptibility to viral clearance among Egyptian patients [44].

Lastly, the Egyptian Genome Project has been recently launched to sequence 100,000 healthy Egyptian adults and 200 ancient Egyptian mummies, including approximately 8000 individuals who may be affected by a genetic condition. This project is expected to contribute to creating the first comprehensive genetic database for Egypt and North African inhabitants. Furthermore, the study aims to determine carrier frequencies of monogenic diseases within the healthy Egyptian population, evaluate the likelihood of disease manifestation, and contribute to the global understanding of these conditions. In addition, analyzing the ancient DNA from Egyptian mummies will investigate historical genetic changes and the underlying genetic factors of specific diseases [45]. Generally, the Egyptian Genome Project represents a significant endeavor that contributes to our understanding of Egyptian genomics, paves the road for precision medicine while shedding light on Egypt’s genetic history, and further adds to the global knowledge of the world about human genomics and precision medicine. Notwithstanding, the collective efforts and initiatives of the scientific community, researchers, stakeholders, and policymakers in the field of PGx in Egypt are still in the early stages of development. Several obstacles hinder the progress of this field in Egypt such as the lack of a polymorphism map for the Egyptian population. Therefore, additional efforts must be made to address and understand the Egyptian genetic variation and its effect on drug responses. 

## 4. Barriers and Challenges Hurdle the Clinical Implementation of PGx in Egypt 

Translating PGx data into actionable interventions in the clinic is the first step toward personalized medicine. Clinical PGx origins could be traced back to 510 BC. A famous example of Pythagoras was when he found that people went sick with fatal hemolytic anemia after consuming beans. Centuries later, we discovered this phenomenon because of a genetic polymorphism in glucose-6-phosphate dehydrogenase (G6PD). This deficiency reduces G6PD activity, resulting in insufficient NADPH production to protect red blood cells against oxidative stress. This triggers acute hemolysis when individuals are exposed to specific oxidizing substances or drugs [46]. Successful implementation of PGx into clinical practice could be achieved through some steps, first adopting a pre-emptive strategy, establishing a monitoring committee, conducting strict quality control, continuously updating the clinical decision support, and lastly, ensuring informative medical education for healthcare professionals (HCPs) [47]. Indeed, roadblocks arise at each step toward PGx translation into clinical practice. These roadblocks manifest in most low- and middle-income (LMICs) countries, such as Egypt. For instance, challenges in scientific research are the most prevalent barrier to the implementation of PGx in Egypt. Other barriers include concerns about the credibility and reliability of tests and the complexity of interpreting research results as it is based on star allele. Currently, scientists are working diligently to create novel and effective yet inexpensive genotyping methods that may be seamlessly integrated with everyday clinical applications. Large-sample multi-center clinical trials are required to develop the evidence base and form a reliable medication guide. Because of the complexity of PGx resulting from the interaction with different targets, real-world research should be relied on to provide results that closely resemble clinical reality and guide clinical decisions [46].

The proper application of PGx relies heavily on information technology. To advance the study and integration of genomic and precision medicine outcomes, it is essential to establish a robust information infrastructure for collecting patient samples and clinical data. Regrettably, in LMICs like Egypt, regional health information systems remain isolated, hindering the sharing and utilization of valuable medical data [46]. In addition, the clinical application of PGx has been hindered by the lack of education and training for HCPs. There is obvious resistance from medical practitioners toward applying PGx-guided drug-prescribing and monitoring decisions, and the proposed reason is the absence of PGx science in the original professional medical curriculum [46]. For example, a study in the UAE, aimed to evaluate healthcare workers’ awareness and perspectives on genetic testing, showed that the most mentioned implementation barrier was the cost of testing, 62%; subsequently, there was a lack of education and a lower percentage of insurance coverage (57.8% and 57.2%, respectively) [48]. This could be tackled by empowering healthcare workers through training and education by incorporating PGx in medical schools to cover the competencies required for this area of practice.

In most drug development studies, determining the right dose is based on Caucasian dosage standards. This means that a notable number of patients would have poor drug responses or serious adverse effects. Domestic data are of huge value; in fact, the lack of reference databases for multiethnic populations such as the Egyptian population urges local drug authorities to vouch for such a commodity and save adverse effects/ineffective dose costs [46]. Many private and public health insurance companies in several countries do not cover PGx testing and treatment including Egypt. The solution to this challenge is to change insurance companies’ attitudes toward PGx testing by increasing reliability and shed light on the cost-effective benefit of PGx testing. Additionally, standard informed consent is required for ethical reasons. Effective informed consent and independent ethical review are essential for protecting the rights of research participants [46].

## 5. Impact of PGx on Drug Development and Drug Discovery

A human gene’s interaction with the drug varies due to external and internal factors. Thus, PGx is essential to develop drugs based on the complexity of humankind. Previous studies have implemented the interactions of the causes on the human genes and their effect; however, some of them fail to locate the predominant factors of human gene response [49]. Although extensive research has been carried out on the single gene with pharmacological interventions, the gene profiles make it difficult to access the whole of the information, and outcomes still do not reach the specific explanation between the gene and other exogenous factors [50]. 

GWAS studies were developed to clarify the relationship between multiple responses of genes and clinical outcomes; however, the lack of reliable statistical significance to allocate the predominant genes and drug reverse is a drawback [49]. To date, whole genome sequencing studies complement the research gap that GWAS studies try to figure out. In addition, the network integration of biomarkers and the entire human genome requires further research that needs more experimental processes to understand the significance of PGx in drug development [51]. Drug development is a continuous process that needs more investigations in the advanced stages due to the less consideration of human gene type and gene–drug targets in the early stages of drug development (Figure 2) [52]. 

Pharmacogenomics and pharmacogenetics, while often used interchangeably, have different roles in different stages of the pharmaceutical drug industry. Pharmacogenetics is considered a part of PGx. Pharmacogenetics deals only with one gene to identify the drug response and dosing in clinical development and drug discovery. Pharmacogenomics should be employed during the first phase of drug development to assess the impact of genes on an individual’s response to drugs at a genome-wide scale. In contrast, pharmacogenetics should be utilized mostly in the later stages of drug development and primarily investigates the impact of unique genetic variations, such as SNPs, on individual therapeutic responses paving the way for more personalized therapy approaches [53,54,55,56,57,58]. Combining both fields in the field of drug development could improve our understanding of the broad and specific genetic factors that influence drug effectiveness. During the early stages, pharmacogenomics can guide the choice of prospective medications and dosage regimens. Pharmacogenetics can enhance patient-specific outcomes by providing a more personalized strategy that not only enhances clinical outcomes but also coincides with the overall goals of precision medicine.

Most of the PGx studies have explored the drug side effects in the late phases and received a high prevalence of confirmed patients who suffer from the drug’s adverse effects; however, these genetic studies may assess the outcome of a specific human gene interaction with a drug rather than figure out the other relevant factors, so this method increases the gap between drug development and human gene response. Recently, a shred of considerable evidence stated that the establishment of PGx needs to be mandatory in the early stages of drug development to improve drug safety and accuracy [52]. The early phase of drug development before drug administration to humans is the best approach to encourage PGx applications and evaluate the human biomarkers related to PK and PD. Humans are different in drug response in the early phase according to the PGx principles, and there are a few study areas that include early evidence of drug effects compared with advanced stages in most pharmaceutical research [1]. It has been noted in some studies that drug development is affected by genetic polymorphism and how humans respond to drugs; for example, during in vitro and in vivo studies, researchers could determine the early target gene responsible for drug toxicity and efficacy, hence enhancing drug efficacy and minimizing drug toxicity [59].

However, in vitro studies cannot predict the relationship between specific drug interactions with gene penalties in mice and humans, such as anion-transporting polypeptide (OATP). In vivo studies are still ongoing to provide data necessary for human studies to investigate human genes from early life to death [60]. In addition, the plasma membrane plays a pivotal role in evaluating drug metabolism. Hence, plasma components are used in the early phase of detecting the antidiabetic drug toxicity by the analysis of anti-histidyl and anti-glycyl-tRNA synthetase antibodies, consequently controlling side effects when the new drug is designed [61]. 

Moreover, considering all the data from multiple in vitro and in vivo studies, it seems that drug development scenarios need to be included in genomic and biomarker studies to ensure the accuracy, safety, and effectiveness of the drug. Yet, this comprehensive approach needs more effort and funding to be applied in the right way to predict the possibility of drug risks [62]. Collectively, studies outline the critical role of drug–gene interaction studies to enhance efficacy and prevent drug events. However, it remains a challenge to find an appropriate approach to integrating all these kinds of data in individualized precision health [63,64,65,66].

Polygenic risk scores (PRSs) are a method that is used to calculate the average weight of multipolar carried risk alleles among the total number of individual alleles by finding the allele relative risk rate. This method tries to predict the possibility of exposed individuals who are already at high-risk compared with non-exposed individuals who have no risks [67]. PRS is a relatively new approach, and many studies have shown its vital role in PGx and drug development [68,69,70,71]. Zhai et al. suggested that a data combination of both GWAS and PGx simultaneously of cardiovascular patients using new PRS showed a successfully applicable approach to predict the accuracy between drug treatment and PGx outcomes among cardiovascular patients [72]. Moreover, it is reported that randomized controlled trials (RCTs) could speed up the image of the perfection of drug and genome biomarker interactions [1]. Sperling et al. (2021) identify that the early phase of atabecestat medication among 557 patients who already have Alzheimer’s disease could prevent the side effects of cognitive functioning in the late phase of drug use. He also recommends that neurological biomarker studies be monitored continuously to avoid the side effects of drug development [73]. 

## 6. PGx of Repurposed Drugs

Recent developments in genetic studies and breakthroughs in understanding the molecular interactions of disease have made precision medicine achievable. The knowledge of approved drugs’ molecular targets and redesigning approved drugs for new indications can be explored. Drug repositioning entails investigating additional uses for a drug previously approved for conditions other than those for which it was initially prescribed. PGx testing of repurposed drugs is necessary to determine therapeutic dosing to avoid or minimize adverse drug reactions [74]. Several criteria, which include drug toxicity, patient co-morbidity, and drug–gene or drug–drug interactions, should be taken into consideration when determining the most effective usage of these repurposed drugs [75]. One significant benefit of repurposed drugs is that they have already undergone safety testing, so the focus can now be on evaluating their effectiveness for a new indication [76]. SARS-CoV-2 is a novel strain of coronavirus [74]. Drug repurposing was utilized in the search for a therapeutic solution for the coronavirus disease (COVID-19) pandemic. The FDA has granted an emergency use authorization for chloroquine to address COVID-19 infection, permitting the use of these medications for unapproved purposes in response to a public health crisis when the pandemic happened, which affected all the countries of the world, including Egypt [77]. 

Antimicrobial agents, non-steroidal anti-inflammatory, cardiovascular, and antidepressant medications are among the potentially repurposed medications in cancer treatment [78,79]. Chloroquine (CQ) and hydroxychloroquine (HCQ) are known antimalarial drugs and are also used in the treatment of SLE and rheumatoid arthritis [80], metabolized by CYP2D6 and CYP2D8 [81], which is expressed differently among individuals. If SNPs occur in *CYP* genes, they may greatly influence CYP enzyme activity. CQ and HCQ are repurposed and used in the management of COVID-19. Their mechanisms of action include the prevention of the virus from entering the host cells by endosomal acidification and inhibiting glycosylation of host ACE2 receptors. They also reversibly inhibit CYP2D6 activity [82]. They may potentiate other drugs metabolized by CYP2D6, and they might cause toxicity of prodrugs that CYP2D6 metabolizes for activation, such as codeine and tramadol [83]. Several variations occur in individuals’ genetic makeup, which could help understand the possible outcomes after specific drug administration. The PK and PD profiles of CQ and HCQ should be assessed to ensure their safety and efficacy in COVID-19 treatment. Genetic variations in CYP enzymes responsible for hepatic metabolism may contribute to individual differences in oral absorption and lead to variations in drug concentrations in the blood and tissues of COVID-19 patients [76]. Extensive PGx testing on repurposed drug trials must be performed to achieve safe and therapeutic dosing and minimize the severity of drug adverse reactions.

## 7. PGx and Pharmacoeconomics

The current economic status of the world draws attention to the pharmaceutical industry, which allows for raising PGx. As studies generate large amounts of “Big Data”, these data processing, analyzing, and interpreting challenges cost money. Nevertheless, the cost of PGx testing itself is high; the main aim of pharmacoeconomics integration into clinical decision-making is to optimize treatment outcomes, minimize adverse reactions, and potentially reduce healthcare costs. PGx screening programs are not widely adopted. The costs of PGx testing vary depending on the particular institution or national healthcare system. Before putting a PGx program into action, it is critical to carefully examine the real-world factors that go into a cost analysis [84]. In any case, PGx creates conflict between the increased money generated by market segmentation and the shift towards a wide range of highly stratified personalized treatments (with commensurately high efficacy) [85]—specifying the cost of the ADRs vs. the total cost of PGx testing and diagnosis is challenging. It differs from one healthcare system to another and from one country to another.

PGx can transform healthcare delivery by enabling personalized medicine, improving treatment outcomes, and resolving the financial issues that healthcare systems worldwide face. Healthcare practitioners can increase the value of healthcare delivery, lower costs, and improve patient care by using genetic information to guide treatment decisions. However, to fully utilize PGx, healthcare stakeholders must work together, invest in infrastructure and research, and pledge to guarantee that all patients have fair access to PGx-guided treatments.

Still, the dilemma of the cost-effectiveness of PGx testing is in its utmost situation, and further research is needed to evaluate it in Egypt. A recent study conducted in the MENA region examined that Pharmacoeconomics implementation is significantly hampered by several factors, including insufficient medical data and its accessibility, a lack of pharmacoeconomic experts, a lack of understanding of the field’s significance, the absence of national governing bodies, ineffective record keeping, and inadequate formulary management [86]. 

Nevertheless, to overcome these barriers, a national effort has to be made to develop specialized regulatory affairs that examine the latest and available PGx testing and conduct local cost-effectiveness to set guidelines for practice and to facilitate the clinical implementation of PGx, and ultimately to have a clear insight into the actual cost-effectiveness of PGx testing, as shown in Figure 3.

## 8. Regulatory Considerations for PGx

Regulatory science is a multidisciplinary field that focuses on developing new tools, methods, and approaches to assess newly innovative drugs [87]. This assessment includes efficacy, safety, quality, dose, and risk–benefit. It balances the need between public health and the life span of the pharmaceutical drug. This includes pre-approval stages of drug development, biomarker identification and validation, and demonstration of clinical efficacy and quality in post-approval pharmacovigilance reports [88]. Two leading agencies regulate the launching of new drugs: the FDA and the European Medicines Agency (EMA) [89]. 

The application of PGx approaches during drug development is an evolving process that begins with discovery and continues to confirm clinical efficacy and safety outcomes. It provides advice on general considerations regarding dating collection and data analysis in early-phase trials. Importantly, it is relevant for observational and exploratory studies intended to generate genomic hypotheses that may be tested in phase 3 trials. It is important to briefly describe the stages required for new drugs in the FDA [90]. Table 1 describes an overview of PGx-related guidance stated by the FDA and the EMA regarding biomarkers, clinical development, drug development, labeling, and required PGx testing through the drug life cycle [91]. Although the FDA states many PGx considerations, several challenges exist, such as reproductivity, moving to practice, and integrating large sets of polymorphic enzymes. On the other hand, the EMA does not give PGx solid consideration. In a review article, the researchers compared CYP450 PGx in both the United States of America (USA) and the European Union (EU). Their findings were (51 vs. 26%) PGx subheadings in the US other than the EU [92,93].

The clinical assessment of PGx stated by the FDA takes into consideration several factors; hence, the FDA is required to fulfill several aspects regarding drug development and PGx biomarkers [44]. In vitro and in vivo studies should assess variations in metabolizing enzymes, transporters, gene variants, and target receptors that might impact PK and PD properties. For example, genetic variation affects drug exposure tests: AUC, C_max_, volumes of distribution, clearance, and half-lives. Hence, drug concentrations in multiple-dose PK studies can give an understanding of appropriate dosing to achieve optimum efficiency in different genotyping.

In addition, clinical pharmacology studies allow the innovator to assess PGx factors such as individual variability. These studies allow researchers to examine the genetic variants that may affect the PK parameters. It also gives information about the toxicity of the investigational drug. For example, to characterize the maximum concentration in oncology studies, it is essential to understand whether high-exposure doses are restricted to the specific subject with particular genotypes. Additionally, single- or multi-dose PK studies in healthy volunteers will be conducted to assess common gene variants with known phenotypes. Another strategy that should be mentioned is routine retrospective genotyping; it means a gene chip that includes different types of genes of metabolisms and transporters of the volunteers. Another consideration for the drugs that are substrates for a polymorphic receptor is that CYP genetic testing is vital for late-phase studies. Polymorphic enzymes include many types: ultra metabolizer (UM), extensive metabolizer (EM), intermediate metabolizer (IM), and poor metabolizer (PM). Thus, PGx helps researchers and innovative companies with (1) optimum patient selection, (2) therapeutic window identification for phase 3, and (3) adverse effects identification.

Dose/response studies are essential in phase 2 to provide proof of concept, dose identification, and dose–response identification for the common adverse effects. These considerations are mainly drug blood levels resulting from genetic variation. Anyway, some genetic markers do not cause an “all or none” response. Hence, including patients with various genetic variations is preferable in the early stages of trials. In addition, taking into consideration several other covariates, such as demographics, and environmental factors, will deepen our understanding of the nature of the interaction between specific genes and other factors that affect the drug response which is essential regarding the impact of genetic and non-genetic factors on PK and PD, dosing, toxicity, and drug safety. 

However, the Egyptian Drug Authority (EDA) does not give a clear point of view regarding PGx’s role in drug approval in Egypt. Some leading pharmaceutical corporations have established PGx testing to initiate the drug. For example, MAYZENT^®^ (Siponimod) requires CYP2C9 genetic testing before initiation by Novartis, which pays for the cost of this investigation testing in Egypt [94]. Another example is Carcemia^®^ (Imatinib), the standard of care for CML. To anticipate variations in clinical responses among Egyptian patients, it is advised to order a PGx testing panel of the following SNPs: *ABCG2*, *ABCB1*, OATP1B3 (*SLCO1B3*), and *CYP3A5* [95]. Briefly, PGx should be considered in the drug approvals in Egypt as it is vital in dosing, identification of side effects, and risk assessment. Additionally, the EDA should change the concept of “One Size Fits All” to deliver the optimum drug efficacy.

**Table 1 pharmaceuticals-17-00881-t001:** Illustrates the aspects of concordance and discordance between the FDA and EMEA in the drug lifecycle: a PGx perspective.

Aspect	FDA	EMA	Ref.
Regulatory Framework	Provides an explicit guide on data submission	Provides a detailed flexible approach on the role of PGx in drug development	[96]
Biomarkers	Requires validation of biomarkers during the process of approval and incorporates them in labeling	Recommends biomarkers use with flexible validation process	[97]
Clinical Development	Recommends the incorporation of PGx data in clinical trials	Recommends the incorporation of PGx data in clinical trials depending on case-by-case clinical situation	[98]
Drug Development	Warrants comprehensive PGx data for new drug applications.	Recommends the incorporation of PGx data, with emphasis on benefit–risk balance	[99]
Labeling	Detailed PGx data encompassed in drug labels to guide personalized prescribing	PGx data included in the drug labels	[100]
PGx Test	Requires PGx testing for drugs with substantial genetic polymorphism	Recommends PGx testing on the basis of evidence, considering data from post-marketing surveillance	[101]

## 9. Legal and Social Implications of PGx

The completion of the human genome project marked a significant milestone in history, leading us to the genomic era and paving the way for PGx to emerge. It focuses on utilizing genomic insights to enhance human well-being in the pharmacy field. Both in research and clinical practice, the applications of PGx strive towards improving health and quality of life. However, despite good intentions, several perspectives among stakeholders in PGx endeavors may lead to conflicting perspectives or actions, thereby giving rise to associated issues [102]. PGx integration into clinical practice brings several legal and social implications that need careful consideration. There are robust necessitates for regulatory frameworks to ensure the safety and effectiveness of genetic testing implementation in clinical settings. Regulatory bodies such as the FDA in the United States and the EMA in Europe are critical in approving PGx tests and guiding their use [103]. 

The introduction of PGx testing raises several questions and doubts about malpractice and liability in cases where adverse drug reactions take place despite genetic testing. Legal clarity is required regarding the responsibilities of healthcare providers in interpreting genetic data, making treatment decisions, and mitigating risks associated with drug–gene interactions. Informed consent is the cornerstone of policy, as genetic testing involves sensitive information about individuals’ genetic ethnicity markers, genetic predispositions, pathological variants, germline information, and potential health risks. Ensuring informed consent is critical to respect patients’ autonomy and protect their privacy rights. Legal frameworks must address data protection, confidentiality, and patients’ rights to access and control genetic information.

Health policies must also be compatible with the patient’s needs and the advancements in the PGx field. Consequently, insurance coverage concerns about the accessibility and affordability of PGx testing need to be addressed and legalized by insurance providers to ensure adequate access to the services and the potential treatments and prevent discrimination based on genetic information, ethnicity, or ancestral information gathered from the genetic testing.

Socioeconomic factors can influence patients’ access to PGx testing and personalized medicine. Addressing healthcare disparities requires strategies to ensure equitable access to everyone across diverse populations, including underserved communities and minority groups. PGx raises ethical dilemmas related to the use of genetic information, such as concerns about genetic discrimination, stigmatization, and the potential misuse of genetic data for non-medical purposes. Societal dialogue and ethical guidelines are essential to navigate these complex issues responsibly. In Egypt, much of the population is fond of religious beliefs and practices; some extrapolated data from such genetic tests will indeed create many questions and concerns. Advancements in genetic testing technologies may intersect with religious beliefs regarding the sanctity of life, family structures, families’ origins, and their origin, religion, or ethnic background. Balancing this versus the potential benefits of genetic testing results and their use in the pharmaceutical industry for disease prevention and treatment should be the highest priority for governments and, in turn, the population.

## 10. Future Perspectives and Opportunities

### 10.1. Integration of PGx Data in Drug Discovery and Development Pipelines

The advancement in gene expression, epigenomics, metabolomics, proteomics, and their application to further understand the molecular basis of diseases and PGx data can impact drug discovery and the developmental pipeline, influencing optimum drug delivery and therapeutic outcomes. Genetic tools are used to identify potential new drug targets and also give insight into the identified targets in the earliest phase of drug discovery. As we progress into the development pipeline, clinical trials at early and late stages have been conducted using PGx to stratify populations.

An example is the drug development for aggressive and lethal treatment-resistant prostate cancer integrating a PGx data-driven approach and PGx-guided computational prediction to identify a nicotinamide phosphoribosyl transferase inhibitor as a top drug candidate potentially effective in the treatment of the most lethal types of prostate cancer [104]. Beyond the application in new drug discovery, PGx data have been utilized to respond to critical questions of toxicity and efficacy or identify potential new indications for developing drugs. PGx use during drug development could translate to a reasonable saving of resources and time in the various phases. It is employed to screen drug compounds and pinpoint potential drug adverse reactions in preclinical drug design before progressing into the elaborate clinical phase.

Pharmacogenetics is also useful in designing drugs by isolating non-metabolizer patient groups. Innovations in high-bit rate genotyping tools improve the facilitation of jumping technical hurdles, clinical trial design, drug development techniques, and circulating drug pharmacovigilance. Pharmacogenetics, thus, enhances drug development at all phases and will fundamentally advance medical practices soon.

### 10.2. The Potential Impact of PGx on Precision Medicine and Healthcare Outcomes

The world has shifted from the ancient approach of one drug fits all to a personalized, individual-oriented treatment approach; Figure 4. What is achievable today is a more personalized, predictive, preventive, participatory approach to healthcare delivery and patient and disease management, though with some challenges but with enormous improvement in healthcare delivery and therapeutics [105].

Data collection from patients through medical diagnosis, individual therapy analysis, and appropriate business designs are required to achieve precision therapy. The role of digital and mobile medical applications in characterizing disease in precision health cannot be overemphasized. Health care at an individual level, beyond genomics, also takes into consideration other information that predicts the risk of disease and the potential outcome of treatment [106].

PGx testing is already applied for selecting and/or dosing selected cardiovascular and psychiatry therapeutic agents [107,108]. In cancer treatment, it is used to identify the genetic predictive and/or prognostic biomarkers linked to drugs to detect and understand genetic polymorphism among individuals, improve therapy efficacy, minimize the risk of toxicity [109], and determine different therapy responses in rheumatoid arthritis patients [110]. PGx has played a vital role in achieving the goal of personalized medicine for many diseases. It has an enormous impact on the prescription of drugs and their dose for many diseases.

## 11. Conclusions

In this review, we delved into the development of PGx, starting from its role and moving toward the current status of PGx in augmenting patient care and drug therapy in Egypt. This article aimed to unravel the intricacies of genetic variation and its influence on drug metabolism, effectiveness, and toxicity in the Egyptian population. Yet, the differentiation of interactions between the Egyptian genome and various drugs is questionable. Because PGx, as a road for personalized medicine, is used for stratifying patients and individualizing dosing regimens, it is a crucial tool for modern clinical practice. PGx, as a road to personalized medicine, is accompanied by roadblocks that must be maneuvered to ensure its translation into practice occurs flawlessly. In Egypt, many national research projects such as ECCO-GEN, Egyptian Systemic Lupus Erythumus, Egyptian Neurodevelopmental studies, and the Egyptian Genome Project are interested in discovering the Egyptian Genome. Subsequently, these initiatives yielded a notable amount of Egyptian data that will allow and encourage clinicians to implement PGx in clinical practice. Notwithstanding, quite an amount of effort, throughout the years, was made to discover the Egyptian Genomic interactions with various drugs. Moreover, the recent economic condition necessitates a vigilant allocation of resources to optimize Egyptian patient outcomes. This review should serve as a lighthouse for national regulatory bodies such as the EDA and the Ministry of Health and Population to develop thick lanes for ultimate PGx implementation across the Egyptian population. Partnerships with technology leaders are vital in inventing cost-effective means of PGx testing. Time will tell how crucial the utilization of PGx is to optimize patient care. In this new era of novel genetic testing, clinicians will shift to personalized care. The highly admixed nature of the Egyptian population necessitates more robust local clinical research studies, both observational and interventional. Large-sample-sized studies are critical to discovering how epigenetic variables and differing individual characteristics indeed share in varying drug responses across different regions in Egypt. Incorporating patients into the decision-making process in patient care needs to be more recognizable and embraced.

## Figures and Tables

**Figure 1 pharmaceuticals-17-00881-f001:**
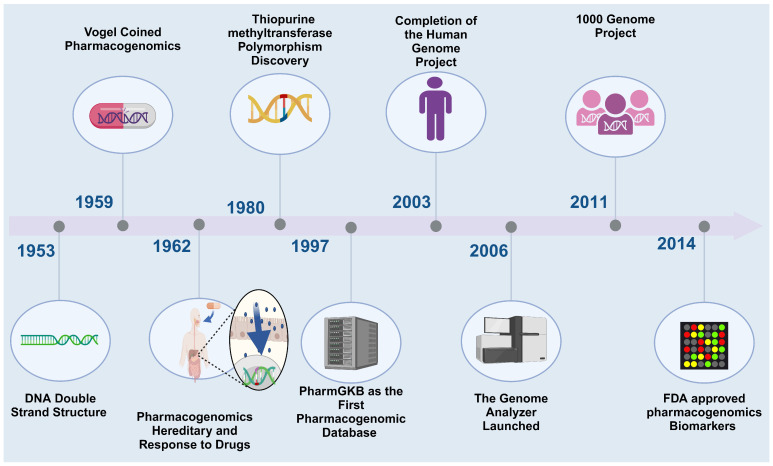
Illustrates the evolving timeline of PGx. Created with BioRender.com (access on 18 January 2024).

**Figure 2 pharmaceuticals-17-00881-f002:**
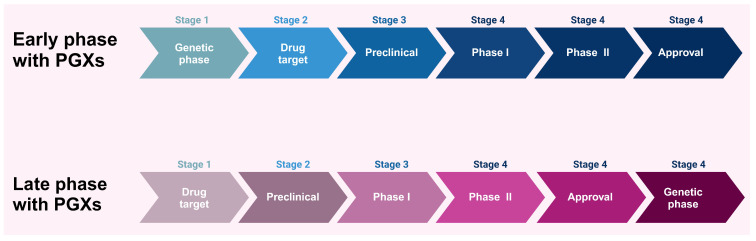
The impact of pharmacogenetics implementation in the early and late stages of the drug development process. Created with BioRender.com (access on 18 January 2024).

**Figure 3 pharmaceuticals-17-00881-f003:**
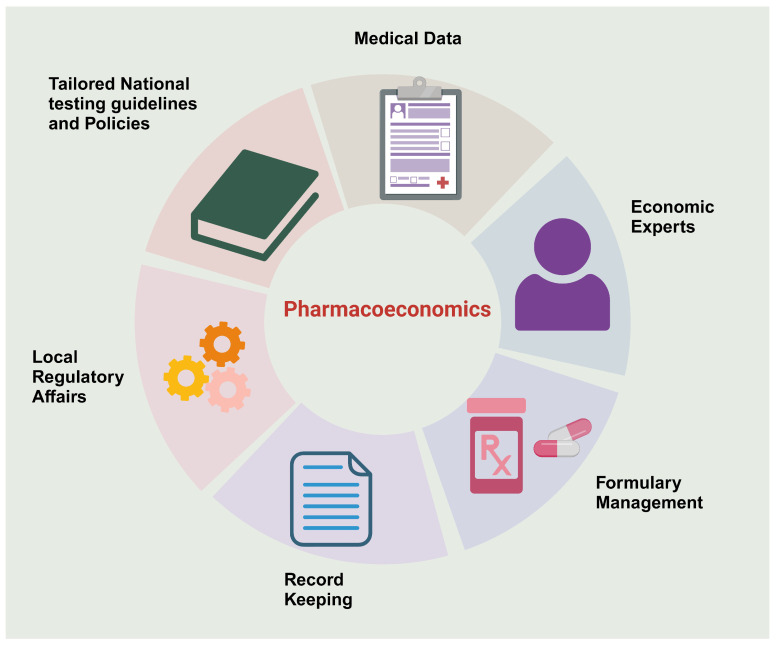
Factors affecting the effective implementation of Pharmacoeconomics to assess the cost-effectiveness of PGx testing. Created with BioRender.com (access on 18 January 2024).

**Figure 4 pharmaceuticals-17-00881-f004:**
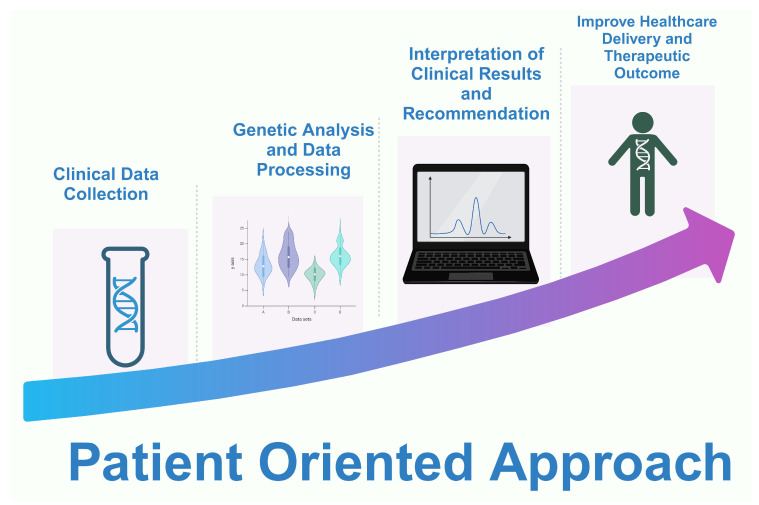
An illustration of different patient-centric approaches in personalized medicine. Created with BioRender.com (access on 18 January 2024).

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
