# Peer review of "From Drug Discovery to Drug Approval: A Comprehensive Review of the Pharmacogenomics Status Quo with a Special Focus on Egypt"

_pharmaceuticals, 2024, doi:10.3390/ph17070881_

Round 1

Reviewer 1 Report

Comments and Suggestions for Authors

This is the revised manuscript (Pharmaceuticals - ISSN 1424-8247) entitled “From Drug Discovery to Drug Approval: A Comprehensive Review of the Pharmacogenomics Status Quo in Egypt” by Abdullah B. Haroun. The authors reviewed the pharmacogenetics current status in research and describes how pharmacogenomic principles are incorporated into clinical practice and in to medication response. The authors concluded that it is very important the collaboration between researchers, physicians, legislators, and government agencies to work together to fully utilize pharmacogenomics' potential to enhance personalized treatment and improve healthcare outcomes.

The topic is of great interest. The manuscript is well-written and clear and the conclusions are well-founded. However, the study could be enhanced by including additional examples of genetic tests that are useful in predicting drug responses, particularly for medications with a high risk of side effects, such as antiviral drugs or anticancer.

Author Response

Reviewer 1:

Comment:

  1. The study could be enhanced by including additional examples of genetic tests useful in predicting drug responses, particularly for medications with a high risk of side effects, such as antiviral or anticancer drugs.
  • As the reviewer suggested, we add additional examples of genetic tests useful in predicting drug responses for antiviral or anticancer drugs (Page 7, lines 285-324)

Reviewer 2 Report

Comments and Suggestions for Authors

The manuscript titled “From Drug Discovery to Drug Approval: A Comprehensive Review of the Pharmacogenomics Status Quo in Egypt” attempted to highlight the status of pharmacogenomics in Egypt.  

However, the manuscript content does not reflect the title. There is very limited data related to Egyptian population. As such, whole manuscript discusses the general pharmacogenomics and lacks any critical evaluation. The objective of this review was not mentioned either in the abstract or in the main text. This reviewer did not find this manuscript aligned with the goal of this work.

Comments on the Quality of English Language

Some correction needed in grammar and syntax.

Author Response

Reviewer 2:

  1. The manuscript content does not reflect the title. There is very limited data related to the Egyptian population. As such, the whole manuscript discusses general pharmacogenomics and lacks any critical evaluation. The objective of this review was not mentioned either in the abstract or in the main text. This reviewer did not find this manuscript aligned with the goal of this work.
  • We would like to thank the reviewer for helping us improve our manuscript, as the reviewer suggested we have changed the title to respect our manuscript aim.
  1. There is very limited data related to the Egyptian population.
  • We are grateful to the reviewer for his valuable comment, in response to the reviewer's comment we extended the section “Current status in Egypt” by further elaborating on initiatives, studies, and the national project to complete an Egyptian reference genome which will help establish the pharmacogenomic database in Egypt. Examples of neurodevelopmental and SLE were mentioned as well in this section.
  • Additionally, in the section on pharmacogenomic considerations in drug regulation, we have mentioned two examples showing PGx testing “Mayzent and Carcemia”. Mayzent requires genotype testing to initiate and determine the dose. Also, Carcemia should have a genotyping, but this is not always the case in Egypt. Although PGx is important in drug regulation, the Egyptian Drug Authority does not consider it.

  1. As such, the whole manuscript discusses general pharmacogenomics and lacks any critical evaluation.
  • In this revised version, we try our best to critically analyze the literature and show where some discrepancies and areas warrant further investigation. In addition, we delivered a critical evaluation in the last few paragraphs of each section.
  1. The objective of this review was not mentioned either in the abstract or in the main text.
  • In response to the reviewer's comments, we stated the objectives and clarified our aim in the abstract section (Page 1, lines 31-40) and introduction section (Page 2, lines 84-89)
  1. This reviewer did not find this manuscript aligned with the goal of this work.
  • We would like to thank the reviewer for his valuable comment, in response to the reviewer's comment we spent much effort to align the manuscript with our intended objective. We revised the abstract and improved the body of the manuscript to enhance the reader's understanding and effectively communicate the originality and implications of our work.

Reviewer 3 Report

Comments and Suggestions for Authors

This manuscript “From Drug Discovery to Drug Approval: A Comprehensive Review of the Pharmacogenomics Status Quo in Egypt” attempts to review the use of pharmacogenomics in the drug development process as well as its regulation, limitations of its use at the clinical level and future perspectives. Interestingly, it also includes the ethical connotations of the use of pharmacogenomics.

There are some aspects that could be improved in the paper:

1)      There are some interesting examples of pharmacogenes but it does not cover the general overview of pharmacogenomics and criteria of choice is not shown. The same for repurposed drugs.

2)      Figures and table are not cited in the text.

3)      There are two figures with number 3.

4)      Heading of the first figure number 3 is not informative.

5)      There are some capital letters inside the text that have not been appropiately used (e.g. line 154, 553). Please, check the text carefully.

6)      Line 172. Are you sure that is it CYP2D5?

7)      Discussion about using pharmacogenomics or pharmacogenetics should be included, since for some drugs there is only evidence of one important SNP in one gene (or only a few of them). Maybe pharmacogenomics is more important in the first stages of drug development and pharmacogenetics in the latter ones?

8)      Please, reconsider to maintain the 12 steps of Drug Approval Processes by the FDA and related figure since they are widely known.

9)      PK and PD or HCP should be explained the first time that they are used.

10)  The implementation of pharmacogenetics/genomics is widely based on the clinical recommendations of CPIC guidelines. Please, add information about it.

Author Response

Reviewer 3:

  1. There are some interesting examples of pharmacogenes but it does not cover the general overview of pharmacogenomics and the criteria of choice are not shown. The same for repurposed drugs.
  • We would like to thank the reviewer for pointing that out. Regarding this comment, a paragraph has been added to explain the selection criteria of the pharmacogenes in the review article, with a brief discussion at the end of this section, highlighting the main discussed points. Another paragraph was added to explain the significance of the selected repurposed drugs in the Egyptian population (Page 4, lines 125-131).
  1. Figures and tables are not cited in the text.
  • As the reviewer suggested, we have cited figures and tables properly across the manuscript.

  1. There are two figures with number 3.
  • We apologize for overlooking this point, in response to the reviewer's comment, we have revised the figure numbering ensuring the correct numbering across the manuscript.

  1. The heading of the first figure number 3 is not informative.
  • In response to the reviewer's comments, we edited the heading of the first figure number 3 properly.

  1. There are some capital letters inside the text that have not been appropriately used (e.g. lines 154, 553). Please, check the text carefully.
  • We are grateful to the reviewer for helping us improve our manuscript. As the reviewer suggested, we have checked typos, capitalization, and grammatical errors across the manuscript.

  1. Line 172. Are you sure that it is CYP2D5?
  • We apologize for overlooking this point, in response to the reviewer's comment, we have replaced the phrase with the correct one “While CYP2D6 has an important role in opioid-related toxicity, CYP2C9 is responsible for the metabolism of anticonvulsant, proton pump inhibitors and antiplatelet such as warfarin”. (Page 5, lines 193-196).
  1. Discussion about using pharmacogenomics or pharmacogenetics should be included, since for some drugs there is only evidence of one important SNP in one gene (or only a few of them). Maybe pharmacogenomics is more important in the first stages of drug development and pharmacogenetics in the latter ones?
  • As the reviewer suggested we emphasized using pharmacogenomics or pharmacogenetics drug development (Page12, lines511-522).
  1. Please, reconsider maintaining the 12 steps of Drug Approval Processes by the FDA and related figures since they are widely known.
  • As the reviewer suggested, we maintained the 12 steps of drug approval processes by the FDA and related figures.

  1. PK and PD or HCP should be explained the first time that they are used.
  • We are grateful to the reviewer for his valuable comment, as suggested we have checked all abbreviations across the manuscript and ensured that they explained the first time that they are used, and only abbreviated form is only used subsequently.

  1. The implementation of pharmacogenetics/genomics is widely based on the clinical recommendations of CPIC guidelines. Please, add information about it
  • In response to the reviewer's comments, we added a piece of more information addressing the clinical implementation of PGx with a scoping focus on Egyptian clinical practice.

Round 2

Reviewer 2 Report

Comments and Suggestions for Authors

Comments were reasonably addressed.

Author Response

The authors would like to thank the reviewer for the valuable support.

Reviewer 3 Report

Comments and Suggestions for Authors

The authors have modified the manuscript according to the suggested proposals but there is still no clear evidence of the clinical utility of this work. Much information is included but the connection between the ideas is poor. Many examples are shown, but not a focused vision to improve the current situation of its implementation in Egypt.

In addition, there is repetitive information in page 4 (120-122) or, again, capital letters are not completely well used (It, line 504).

Author Response

Point to point response to the reviewers’ comments

We are grateful to the reviewers for helping us improve our manuscript. Here is our point-to-point response to the reviewers’ comments.  

  1. The authors have modified the manuscript according to the suggested proposals but there is still no clear evidence of the clinical utility of this work.
  • We would like to thank the reviewer for acknowledging our work
  1. Much information is included but the connection between the ideas is poor. Many examples are shown, but not a focused vision to improve the current situation of its implementation in Egypt.
  • We are grateful for the reviewer’s valuable comment, in response to the reviewer’s comment we restructured our manuscript to increase the coherence and logical flow of the ideas, make connections when possible, and finally, remove any/all of the unnecessary information without affecting the context. Finally, we dissected the challenges and the barriers that hinder the implementation of PGx in Egypt and we provide possible solutions that improve the current situation of its implementation in Egypt.
  1. In addition, there is repetitive information in page 4 (120-122)
  • In response to the reviewer’s comment, we removed repetitive information cross the manuscript.

  1. again, capital letters are not completely well used (It, line 504).
  • We apologize for overlooking this point, as suggested we checked the review for improper capitalization, typos, and inconsistencies.

Round 3

Reviewer 3 Report

Comments and Suggestions for Authors

The authors have substantially improved the paper.